# High temperature superconductivity at FeSe/LaFeO$_3$ interface

Yuanhe Song[1], Zheng Chen[2], Qinghua Zhang[3], Haichao Xu [1,4], Xia Lou[1], Xiaoyang Chen[1], Xiaofeng Xu[3], Xuetao Zhu[3], Ran Tao[1], Tianlun Yu [1], Hao Ru[1], Yihua Wang[1,4], Tong Zhang[1,4], Jiandong Guo[3 ✉], Lin Gu [3 ✉], Yanwu Xie [2 ✉], Rui Peng [1,4 ✉] & Donglai Feng [4,5,6 ✉]

Enormous enhancement of superconducting pairing temperature ($T_g$) to 65 K in FeSe/SrTiO$_3$ has made it a spotlight. Despite the effort of interfacial engineering, FeSe interfaced with TiO$_x$ remains the unique case in hosting high $T_g$, hindering a decisive understanding on the general mechanism and ways to further improving $T_g$. Here we constructed a new high-$T_g$ interface, single-layer FeSe interfaced with FeO$_x$-terminated LaFeO$_3$. Large superconducting gap and diamagnetic response evidence that the superconducting pairing can emerge near 80 K, highest amongst all-known interfacial superconductors. Combining various techniques, we reveal interfacial charge transfer and strong interfacial electron-phonon coupling (EPC) in FeSe/LaFeO$_3$, showing that the cooperative pairing mechanism works beyond FeSe-TiO$_x$. Intriguingly, the stronger interfacial EPC than that in FeSe/SrTiO$_3$ is likely induced by the stronger interfacial bonding in FeSe/LaFeO$_3$, and can explain the higher $T_g$ according to recent theoretical calculations, pointing out a workable route in designing new interfaces to achieve higher $T_g$.

[1] Laboratory of Advanced Materials, State Key Laboratory of Surface Physics, and Department of Physics, Fudan University, 200438 Shanghai, China. [2] Department of Physics, Zhejiang University, 310027 Hangzhou, China. [3] Beijing National Laboratory for Condensed Matter Physics and Institute of Physics, Chinese Academy of Sciences, 100190 Beijing, China. [4] Shanghai Research Center for Quantum Sciences, 201315 Shanghai, China. [5] Hefei National Laboratory for Physical Science at Microscale and Department of Physics, University of Science and Technology of China, 230026 Hefei, Anhui, China. [6] Collaborative Innovation Center of Advanced Microstructures, 210093 Nanjing, China. ✉email: jdguo@iphy.ac.cn; l.gu@iphy.ac.cn; ywxie@zju.edu.cn; pengrui@fudan.edu.cn; dlfeng@ustc.edu.cn

The discovery of high-temperature superconductivity at FeSe/SrTiO$_3$ interface has ignited intensive research interests[1-8]. The superconducting gap opening temperature, which signals the formation of Cooper pairs ($T_g$), is as high as 65 K in FeSe/SrTiO$_3$, and further reaches 75 K after tuning the tensile strain and correlation strength in FeSe/BaTiO$_3$ (Peng et al.[6]). Although the superconducting coherence temperature is still debated[7-13], it is generally agreed that FeSe interfacing with TiO$_x$-terminated oxides (noted as FeSe-TiO$_x$ hereafter) holds the highest pairing temperature among Fe-based superconductors and monolayer films. This is in stark contrast to the diminishing superconductivity of single-unit-cell (1uc) FeSe on graphene[14], and also significantly higher than the superconducting transition temperature ($T_c$) of electron-doped FeSe ($e$-FeSe) without an oxide interface[6,15], demonstrating the crucial role of the oxide interface.

Intensive studies have been devoted to elucidate the role of the FeSe-TiO$_x$ interface[5,6,15-33]. Recent resonant inelastic x-ray scattering (RIXS) studies suggest that the magnetic excitation of FeSe/STO is different from its bulk counterpart[34]. Interfacial charge transfer and strain effect have been identified, but are insufficient to account for the high $T_g$ (refs. [6,15,16]). Interfacial electron-phonon coupling (EPC) is suggested based on the observation of replica bands, which could help superconductivity[5,20,21]. Although the replica bands are later questioned and proposed to be extrinsic[18], recent angle-resolved photoemission spectroscopy (ARPES) studies disfavor this proposal and support its intrinsic relation with interfacial EPC (refs. [19,31]). Intriguingly, the superconducting gap is found to scale linearly with the interfacial EPC strength with a finite intercept at zero coupling limit[19]. These results support the scenario that the interfacial EPC cooperates with the spin fluctuation in $e$-FeSe itself in inducing such high $T_g$'s (refs. [5,19-30]). However, how the interfacial EPC and spin fluctuations cooperate remains debated[32,33]. If the mechanism is indeed working, it should happen at other interfaces generally[35]. Experimentally, no successful pairing enhancement over $e$-FeSe has been reported at interfaces beyond FeSe-TiO$_x$. This not only limits the interfacial engineering for higher $T_g$, but also hinders the elucidation of a general picture of interfacial superconductivity.

Here we constructed a new interface of FeSe and FeO$_x$ (noted as FeSe–FeO$_x$ hereafter) with high quality, by epitaxially growing FeSe on 6 unit cells (uc) LaFeO$_3$ (LFO)/Nb:SrTiO$_3$ (STO) heterostructures. The thickness of LFO is chosen to be 6uc because it is thick enough to prevent any intermixing of TiO$_x$ to the FeSe–FeO$_x$ interface, and is thin enough to avoid the photoemission charging effect in the ARPES studies. The $T_g$ measured by in-situ ARPES and the pronounced diamagnetism measured by mutual inductance on the same sample both demonstrate superconducting pairing above the optimal $T_c$ of $e$-FeSe. We show that the highest-achieved superconducting pairing is up to 80 K, much higher than 65 K of FeSe/STO. Electronic structure and phonon spectrum measurements suggest that FeSe/LFO shares the similar cooperative mechanism of superconductivity enhancement as in FeSe-TiO$_x$, but hosts a stronger interfacial EPC. The stronger interfacial EPC in FeSe/LFO almost quantitatively accounts for its higher $T_g$ based on a recent quantum Monte–Carlo simulations[28]. The shorter interfacial bond length at FeSe/LFO should be responsible for stronger interfacial EPC, which points out a route of designing new materials with stronger interfacial bonding for further enhancing $T_g$.

by scanning transmission electron microscopy (STEM). As shown in Fig. 1a, the first uc FeSe is atomically resolved, which remains stoichiometric without intermixing from the capped Se, in contrast to the blurred structure of the additional 0.5uc FeSe. Based on the element-resolved maps (Fig. 1b), the interdiffusion between Ti and Fe is within ±2uc at LFO/STO interface, while FeSe is clearly interfaced with FeO$_x$ rather than TiO$_x$. Two additional FeO$_x$ layers are observed at the interface (Fig. 1c–e) originated from surface reconstruction of LFO. This is similar to the additional TiO$_x$ layer observed at FeSe/STO interface due to the surface reconstruction in STO (refs. [36-39]). Considering the lattice mismatch between bulk LFO ($a \sim 3.93$Å) and STO ($a \sim 3.905$Å) is only 0.64%, epitaxial LFO is expected to follow the in-plane lattice of STO. Consistently, from large scale STEM image (Supplementary Fig. 2), the in-plane lattice of FeSe and 6uc LFO well matches that of the STO substrate. Therefore, the strain effect on FeSe in FeSe/LFO/STO is identical to the well-studied FeSe/STO. The interlayer distances derived from STEM images are indicated in Fig. 1e (see Supplementary Note 2 for details). The anion height and the in-plane lattice of FeSe are both similar to those in FeSe/STO (Peng et al.[37]), therefore the bond angle of FeSe layer remain similar with respect to FeSe/STO. The distance between FeSe and top FeO$_x$ layers is ~2.5 Å, which is smaller than the 2.9 Å distance between FeSe and top TiO$_x$ in FeSe/STO (Peng et al.[37]), reflecting a stronger bonding at FeSe/LFO interface than FeSe/STO considering that Fe and Ti have similar ionic radii.

To understand the electronic behavior and interfacial interactions of this new interface, in-situ ARPES studies were performed. Note that high-resolution ARPES studies require perfect grounding of the sample surface, however, highly insulating LaFeO$_3$ films pose severer challenges to the ARPES measurements. To facilitate the grounding, gold is sputtered at the LFO/STO edge and then covered with silver paste before FeSe growth (Fig. 2a). Photoemission charging effect is carefully inspected (see Supplementary Note 3), and spectra without charging effect can be obtained with reduced photon flux when LFO is 6uc. The in-plane photoemission intensity map at Fermi energy ($E_F$) on 1uc FeSe/LFO/STO (Fig. 2b) shows two elliptical electron pockets located at the Brillouin zone corner. The absence of hole Fermi surfaces indicates that the 1uc FeSe is heavily electron-doped by the interface. Its doping level derived from the Fermi surface volume according to Luttinger theorem is 0.087 $e^-$ per Fe, slightly less than the optimal doping of 0.12 $e^-$ in the electron-doped FeSe (Wen et al.[15]). From the photoemission spectra along high symmetric cuts #1 and #2 (Fig. 2c–f), two parabolic bands (noted as $\alpha$ and $\beta$) can be observed near the $\Gamma$ point with the band top at $-71$ meV, while two electron bands (noted as $\gamma1$ and $\gamma2$) are clearly resolved around the M point with the band bottom at $-50$ meV. The band structure is quite similar to that of FeSe/STO except for a 10 meV downward shift of chemical potential, consistent with the slightly lower electron doping in single-layer FeSe/LFO. It is worth noting that $\alpha$, $\beta$ at $\Gamma$, and $\gamma1$, $\gamma2$ at M are from different orbitals[40] and present in bulk materials (refs. [15,41,42]). Due to the small energy/momentum separation between them, the clear identification of each band under the same experimental setup demonstrates high quality of the samples with minimal impurity scattering (see Supplementary Fig. 8). The high quality of the interface and low photoemission background allow the observation of band replicas ($\alpha'$, $\beta'$, $\gamma1'$, and $\gamma2'$, Fig. 2c–h), which reflect the existence of interfacial EPC (Song et al.[19]).

## Results

**Interfacial atomic and electronic structure**. Figure 1 shows the cross-sectional atomic structure of an amorphous-Se-capped (noted as $a$-Se hereafter) 1.5uc FeSe on 6uc LFO/STO identified

**Enhanced superconductivity**. From the surface morphology measured by in-situ scanning tunneling microscopy (STM), the surface coverage of FeSe is slightly less than the nominal amount

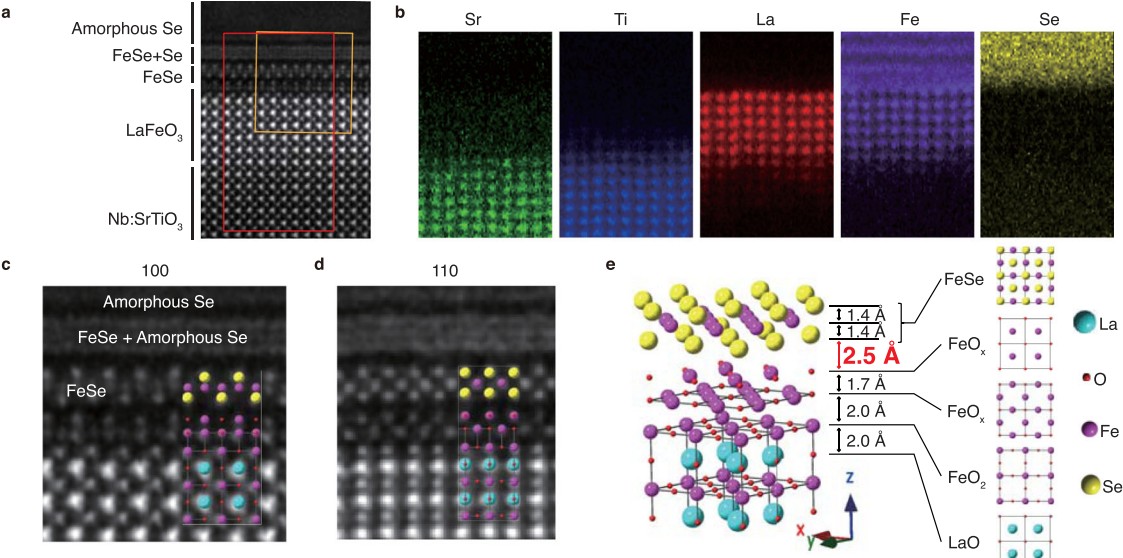

**Fig. 1 Interfacial atomic structure of *a*-Se capped 1.5uc FeSe/LFO/STO. a** High-angle annular dark field (HAADF) image of amorphous Se capped 1.5uc FeSe/LFO/STO along the [100] direction of the STO lattice. **b** Element-resolved maps based on the EELS data at Sr-$L_{2,3}$, Ti-$L_{2,3}$, La-$M_{4,5}$, Fe-$L_{2,3}$, and Se-$L_{2,3}$ absorption edges. The region is shown by the red rectangle in panel **a**. FeSe layer that is atomically resolved in HAADF-STEM loses atomic resolution in element maps due to the dose damage from the much more intense electron beam used during element maps. **c** Zoomed-in HAADF image near the FeSe/ LFO interface, and the side view of the atomic structure is illustrated. The region is shown by the orange rectangle in panel **a**, **d** Same as panel **c**, but along [110] direction of the STO lattice. **e** Illustration of the resolved interfacial atomic structure at FeSe/LFO interface (see Supplementary Note 2 for some details on analysis).

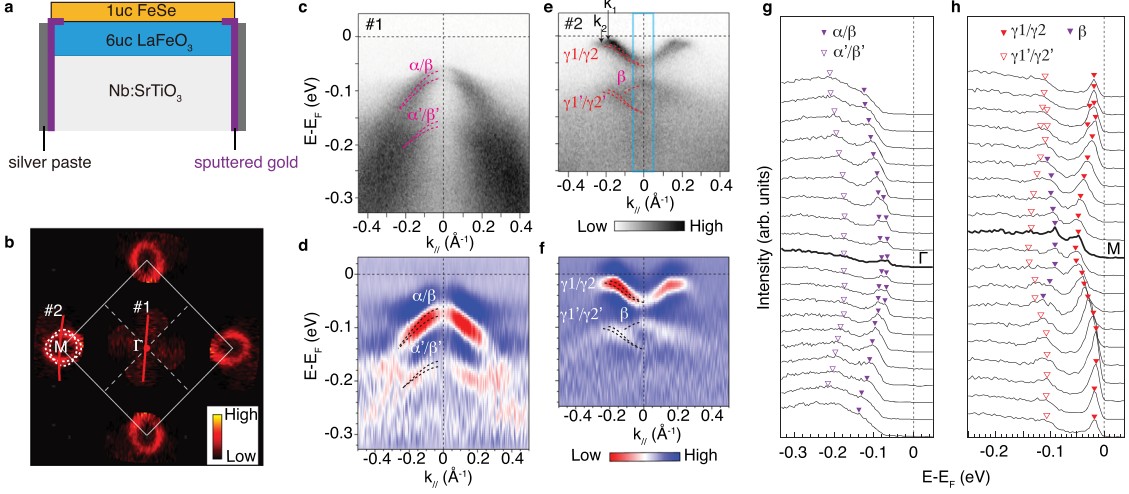

**Fig. 2 Electronic structure of 1uc FeSe/LFO/STO. a** Schematic illustration of the heterostructure. **b** Photoemission intensity map for 1uc FeSe/LFO/STO integrated over ($E_F$−35 meV, $E_F$+15 meV). Data were measured at 6K with full photon flux of the Helium lamp, which shows a photoemission charging effect of ~10 meV (see Supplementary Fig. 5). With the integration energy window of ($E_F$−35 meV, $E_F$+15 meV), it can represent the Fermi surface map under no charging effect. **c**, **d** Photoemission intensity (**c**), second derivatives of the photoemission intensity with respect to energy (**d**) along cut #1 in **b**. **e**, **f** Same as **c**, **d**, but along cut #2 in **b**. The dashed curves trace the band dispersions. **g** Energy distribution curves (EDCs) across Γ along cut #1. **h** Same as panel **g**, but across M along cut #2. Data in **b**–**h** were measured at 6K with 1/8 of the total photon flux of the Helium lamp, which eliminates the photoemission charging effect (see Supplementary Note 3).

of 1uc (Fig. 3a). The 1uc FeSe films are weakly interconnected on each LFO terrace, which leads to poor grounding and charging effect during photoemission measurements. The spectra of 1uc FeSe/LFO/STO were taken using reduced photon flux to eliminate charging effect (Supplementary Note 3). Figure 3b shows the symmetrized photoemission spectrum of 1uc FeSe/LFO/STO around M point at 6 K. The two electron bands γ1 and γ2 backbend near $E_F$, a hallmark of the Bogliubov quasiparticle dispersion. The superconducting gap at $k_1$ ($\Delta_1$) of 1uc FeSe/LFO/ STO is 17 ± 2 meV. The 17 meV superconducting gap is

repeatable on another 1uc FeSe/LFO/STO sample under the measurement condition without photoemission charging effect (Supplementary Note 4). Note that the superconducting peak is sharp (see Supplementary Fig. 7 for detailed sample quality comparison), and the fitted $\Gamma_1$ representing the single-particle scattering rate (20 meV) is comparable to that in high quality FeSe/STO samples[19]. In this case, the extrinsic broadening effect on the superconducting coherence peak is minor, and the gap size, which represents the superconducting pairing strength, can be compared with that of FeSe/STO in Song et al.[19] with same $\Gamma_1$

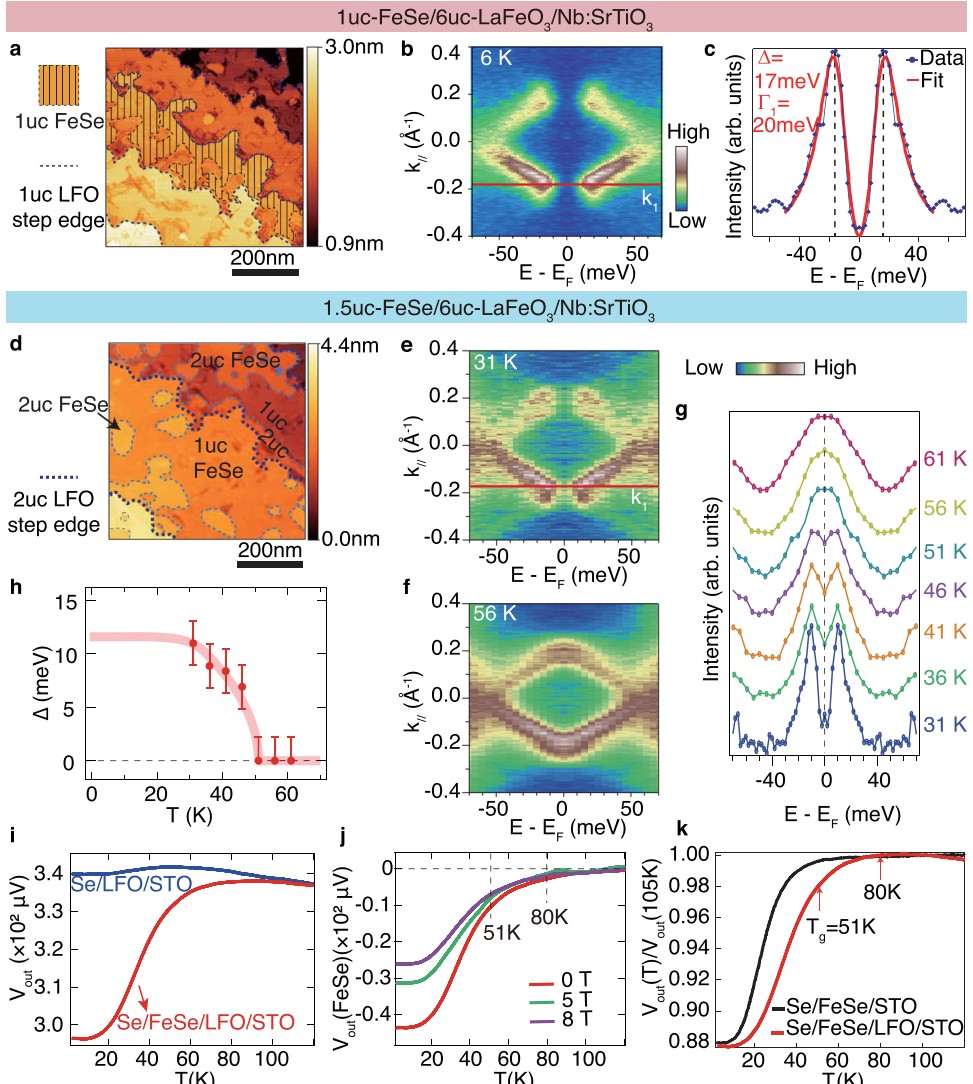

**Fig. 3 Superconducting gap and mutual inductance measurements on FeSe/LFO/STO. a, d** Surface morphology of the nominal 1uc FeSe/LFO/STO (**a**, sample bias = 4 V, tunneling current = 30 pA), and 1.5uc FeSe/LFO/STO sample (**d** sample bias = 2 V, tunneling current = 30 pA) measured by STM. Partial coverage of FeSe on one LFO terrace is illustrated as shaded area in panel **a**. **b** Symmetrized photoemission spectrum with respect to $E_F$ of the 1uc FeSe/LFO/STO across M measured with 1/8 $I_0$ at 6 K. **c** Symmetrized EDC at the $k_1$'s of 1uc FeSe/LFO/STO and the fitting to a superconducting spectral function based on the simplified Bardeen-Cooper-Schrieffer(BCS) self-energy[19,60–62]. **e, f** Symmetrized photoemission spectrum with respect to $E_F$ of the 1.5uc FeSe/LFO/STO across M measured at 31 and 56 K, respectively. **g, h** Temperature dependence of the symmetrized EDC at the $k_1$', and the determined superconducting gap of 1.5uc FeSe/LFO/STO, respectively. The error bars of gap are from the s.d. of the fitting process and the measurement uncertainty. **i** Temperature dependence of the out-of-phase voltage $V_{out}$ in the pickup coil measured by the ex-situ mutual inductance experiments on the a-Se capped 1.5uc FeSe/LFO/STO and a comparative sample of Se capped LFO/STO with the same heat treatment. **j** Diamagnetic signal of FeSe in varied magnetic field. $V_{out}$(FeSe) is obtained by the subtraction of $V_{out}$(Se/1.5uc FeSe/LFO/STO) by $V_{out}$(Se/LFO/STO) in **i**. **k** Comparison between $V_{out}$(Se/1.5uc FeSe/LFO/STO) in this work and $V_{out}$(Se/2uc FeSe/2uc (Fe$_{0.96}$Co$_{0.04}$)Se/1uc FeSe/STO) in Zhang et al.[7] normalized by the corresponding 105K data.

(see Supplementary Note 5 for details). The superconducting gap $\Delta_1$ of 1uc FeSe/LFO/STO (17 ± 2 meV) is larger than those of 1uc FeSe/STO (10–12.1 meV) measured by ARPES with identical sample quality, indicating stronger Cooper pairing strength in FeSe interfaced with LFO. Based on the approximately linear relation between $T_g$'s and superconducting gaps measured at low temperatures[19], the pairing temperature of 1uc FeSe/LFO/STO can be up to 89 K ± 10 K.

Due to the extremely low photon flux used and the subsequent long data acquisition time, it is not feasible to take a complete data set of temperature dependent gap evolution on 1uc FeSe/LFO/STO before the sample is aged. To improve the interconnection of 1uc FeSe and enable photoemission studies at

higher photon flux, we grew an additional nominal amount of 0.5uc FeSe on this sample. The surface of the 1.5uc sample show well-connected 1uc FeSe and several 2uc FeSe patches (Fig. 3d). The photoemission signal from 1uc FeSe can be well distinguished from that from the 2nd uc FeSe, which is without much interfacial charge transfer (see Supplementary Note 6). By analyzing the signal of bare 1uc FeSe in the sample, we can observe a superconducting gap $\Delta_1$ of 11 meV at 31 K (Fig. 3e, g). The gap size is reduced from 17 meV before the growth of the additional 0.5uc FeSe, although they were both determined from the 1uc FeSe signal. This is not surprising considering the delicate superconductivity in 1uc FeSe which is easily affected by slightly different growth and annealing conditions[19,37,43]. The

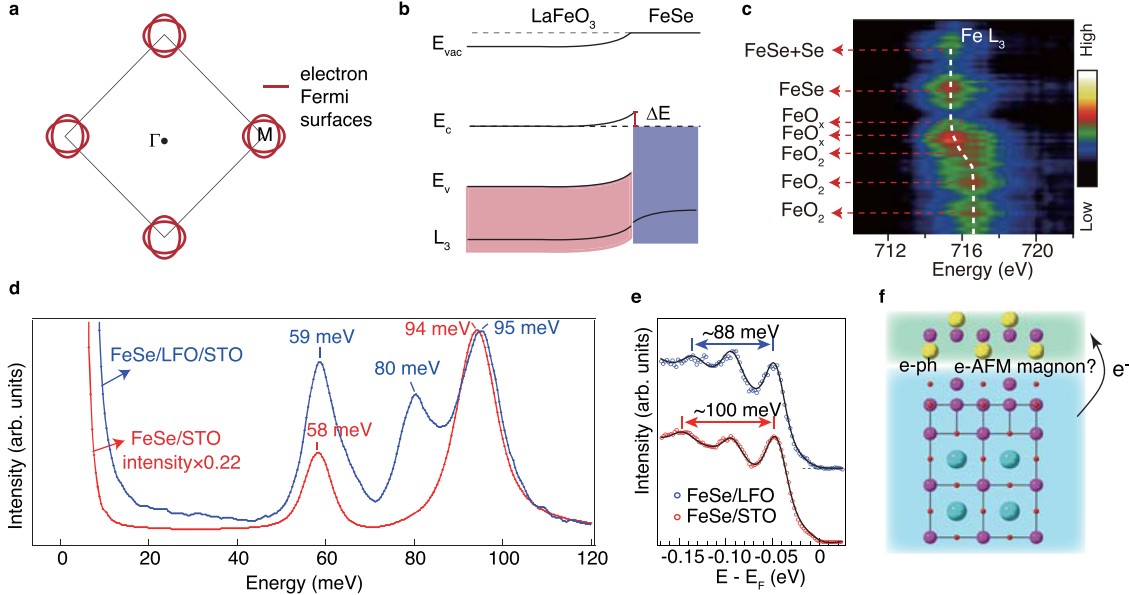

**Fig. 4 Interfacial charge transfer and electron–phonon interactions. a** Illustration of the measured Fermi surfaces of 1uc FeSe/LFO/STO. The Fermi surfaces consist of only electron pockets without hole pockets. **b** Sketch of the work function difference and band bending. **c** Core level EELS map near Fe $L_3$ edge across the interface for the 1.5uc FeSe/LFO/STO measured in Fig. 1. The dash lines mark the peak positions of the individual EELS spectrum. **d** High-resolution electron energy loss spectra of the 1.5uc FeSe/LFO/STO, which is compared with the data on 1uc FeSe/STO (Song et al.[19]). After the total intensity is normalized by the elastic peak, the red line is scaled by a factor of 0.22. **e** EDCs at M of FeSe/LFO and FeSe/STO, which are integrated over the momentum range indicated by blue box in Fig. 2d. The data of FeSe/STO in panels **d** and **e** are reproduced from Song et al.[19]. **f** Sketch of the interfacial interactions which can help enhancing the superconductivity at FeSe/LFO interface.

superconducting gap decreases with increasing temperature (Fig. 3g), and eventually closes above 51 K following BCS formula (Fig. 3h). $T_g$ of 51 K is higher than that of $e$-FeSe with the same doping level of 0.087 $e^-$/Fe ($T_g \sim 40$ K) (Wen et al.[15]).

The superconductivity of the 1.5uc FeSe sample was further characterized by ex-situ mutual inductance measurements on the sample protected by $a$-Se capping layers. The additional 0.5uc FeSe layer protects the underneath 1uc FeSe from direct contact with the Se capping and preserve its stoichiometry (Fig. 1), which is crucial for ex-situ superconductivity measurements. Figure 3i shows the temperature dependence of the out-of-phase voltage $V_{out}$ in the pickup coil measured on the sample. Plotted together is the signal from an $a$-Se capped 6uc LFO/STO processed with the same heat-treatment procedure. Diamagnetic signal is observed in $a$-Se/FeSe/LFO/STO, while absent in $a$-Se/LFO/STO. By subtracting the signal from the $a$-Se/LFO/STO which represents the background inductance from both the coils and the substrate, we can see that the $V_{out}$ of FeSe deviates from zero around 80 K, drops rapidly below 51 K, and eventually saturates around 10 K (Fig. 3j). The diamagnetic signal is suppressed by magnetic field (Fig. 3j), in line with the typical behavior of two dimensional superconductors[1,7,13]. Compared with previous mutual inductance study that suggests $T_c \sim 65$ K in FeSe/STO with both Se and FeSe cappings (Zhang et al.[7]), the diamagnetism occurs at higher temperature in FeSe on LFO (Fig. 3k). These results imply that FeSe/LFO potentially breaks the pairing temperature record of FeSe on titanates.

## Discussion

To reconcile the observation of $T_g \sim 51$ K and the two diamagnetic response at 51 K and 80 K, the delicate and inhomogeneous nature of superconductivity in 1uc FeSe should be considered, similar to that in FeSe/STO interface[7,43]. The dramatic drop in $V_{out}$ (Fig. 3k) happens near $T_g$, suggesting that most portion of the sample shows a pairing temperature of 51 K. On the other

hand, from 80 K to 51 K, there is 3% drop in $V_{out}$, suggesting that a small portion forms superconducting pairs at a temperature as high as 80 K. A partial gap opening is expected between 51 K and 80 K if there is a small portion in the sample possessing $T_g \sim 80$K, but it is hard to observe due to the much higher intensity of normal state spectra (see simulations in Supplementary Note 7 for details). The diamagnetic drop at the 1.5uc sample and the 17 meV superconducting gap in the 1uc FeSe sample both suggest that the highest achievable pairing temperature of 1uc FeSe/LFO can be up to 80 K. Improving the quality of LFO film with pure FeO$_x$ termination could be helpful for increasing the homogeneity and $T_g \sim 80$K regions, which calls for future work on tuning the pulsed laser deposition (PLD) growth parameters or using the oxide molecular beam epitaxy (MBE) technique to grow LFO.

So far superconductivity surpassing optimally doped bulk FeSe has been observed in FeSe/STO, FeSe/BaTiO$_3$, FeSe/TiO$_2$ heterostructures[2–6,16,17,44], all interfaced with the TiO$_x$ terminated substrates. Interfacial EPC in addition to charge doping is generally observed in these FeSe-TiO$_x$ terminated interfaces, in accordance with the cooperative mechanism of interfacial high-temperature superconductivity. Although FeSe has been grown on MgO and NdGaO$_3$ (NGO) substrates, the $T_c$'s [18 K for FeSe/MgO (Zhou et al.[45]) and 28 K for FeSe/NGO (Yang et al.[46])] are lower than that of optimally doped bulk FeSe (Wen et al.[15]), while the interfacial effect is unclear due to the lack of electronic structure study. Here in FeSe/LFO, higher superconducting pairing temperature is observed, while the systematic electronic structure studies offer an ideal test bed for the mechanism of interfacial high-temperature superconductivity.

Heavy electron doping up to 0.087 $e^-$ per Fe is observed in FeSe/LFO/STO by the direct measurement of Fermi surface (Fig. 4a). In FeSe/STO, the work function difference between FeSe and STO is proposed to drive the electrons to FeSe (refs. [36,47]), and the charge transfer with finite screening length induces a bending in the core level of Fe L edge[47]. According to previous reports on bulk materials or thick films, the work function of LFO

$[\phi(LFO) = 4.6\,\text{eV}]$ (Hong et al.[48]) is close to that of STO $[\phi(STO) = 4.5\,\text{eV}]$ (Zhang et al.[47]), while smaller than those of FeSe $[\phi(FeSe) = 5.1\,\text{eV}]$ (Zhang et al.[47]) and MgO $[\phi(MgO) = 4.94\,\text{eV}]$ (Lim et al.[49]). Such work function difference in FeSe/LFO interface would give rise to a band bending similar to that in FeSe/STO (Fig. 4b), which should be much larger than that in FeSe/MgO. Consistently, by measuring the cross-sectional electron energy loss spectroscopy (EELS) spectrum with the incident electron energy of 200 keV (Fig. 4c), a red shift of Fe $L_3$ edge is observed from LFO to the interfacial FeSe. These results suggest that the interfacial charge transfer in FeSe/LFO can be qualitatively accounted for by the work function mismatch scenario similar to FeSe/STO (refs. [36,47]), while a quantitative comparison call for a precise determination on the work function of each ingredient in the heterostructure. The red shift at LFO side can be alternatively explained as a self-reconstructed reduction of Fe valence at the interfacial $FeO_x$ layers. In this case, electrons accumulated at the top $FeO_x$ layers can serve as a charge reservoir to single-layer FeSe, which would facilitate the interfacial charge transfer originated from work function mismatch. In the absence of interface, $0.087\,e^-$ doping per Fe would enhance the $T_g$ to $\sim 40\,\text{K}$ for FeSe (Wen et al.[15]), which is not sufficient to explain the $T_g$ in FeSe-LFO interface.

Replica bands are clearly observed (Fig. 2), indicating the existence of interfacial EPC. To understand the phonon modes, reflective high-resolution EELS (HREELS) with the electron energy of 110 eV was conducted on the surface of the 1.5uc FeSe/LFO/STO sample. Compared with HREELS study on 1uc FeSe/STO, an additional 80 meV peak is observed, originated from the longitudinal optical phonon of LFO with the motions of oxygen atoms relative to Fe (Jamil et al.[50]). The peak at 95 meV originates from STO phonon, and its intensity is 22% of the STO phonon in FeSe/STO (note that the red line is scaled by 0.22 in Fig. 4d), indicating that the electric field generated by STO Fuchs-Kliewer phonons is partially reduced by the LFO layers[51]. The 59 meV peak is contributed by both LFO and STO phonons[50,51]. In-situ ARPES show band replica behavior with the energy separation ($E_S$) of 88 meV (Figs. 2d, f and 4e), larger than the LFO phonon energy of 80 meV due to band renormalization from interfacial EPC, consistent with an intrinsic origin of interfacial EPC (Li et al.[28]). Despite that the electric field generated by the 95 meV STO phonons is present in HREELS, the corresponding replica band ($E_S \sim 100\,\text{meV}$ in FeSe/STO) is absent or of much weaker intensity (Fig. 4e) (refs. [5,17,19]), indicating the STO-originated interfacial EPC is significantly reduced due to the 6 uc of LFO between FeSe and STO.

The band renormalization from interfacial EPC, quantified by $E_S/\Omega$ ($\Omega$ representing corresponding phonon energy), is positively correlated with interfacial EPC constant $\lambda$ (refs. [23,28]). Directly from the raw data without any assumption on spectral background, $E_S/\Omega$ at FeSe/LFO interface is determined to be 88 meV/80 meV = 1.1, which is larger than the $E_S/\Omega \sim 100\,\text{meV}/94\,\text{meV}=1.06$ in FeSe/STO interface[19]. According to a recent quantum Monte–Carlo simulation[28], $E_S/\Omega$ of 1.1 and 1.06 correspond to interfacial EPC constant $\lambda$ of approximately 0.5 and 0.3, respectively. This implies that the interfacial EPC in FeSe/LFO is larger than that in FeSe/STO by nearly 70%, which would enhance the superconducting pairing accordingly[28]. Consistently, the determined intensity ratio of replica band $\beta'$ to the main band $\beta$ is between 0.40 and 0.52 (see Supplementary Note 8), clearly larger than that in FeSe/STO ($\sim 0.2$) (Song et al.[19]). This is qualitatively consistent with the larger EPC strength determined according to the blue shift of the energy separation relative to the phonon energy. As the $T_g$ in FeSe/STO is enhanced by $\sim 20\,\text{K}$ from $e$-FeSe, assuming a linear relation between the gap enhancement and $\lambda$ (refs. [19,23]), the stronger

interfacial EPC in FeSe/LFO is expected to give an enhancement of $\sim 34\,\text{K}$ in addition to the $T_g \sim 40\,\text{K}$ of $e$-FeSe (Li et al.[28]), which roughly explains the $T_g$ up to 80 K in FeSe/LFO. The nearly quantitative agreement between our experimental results and the simulation in Li et al.[28] follows the cooperative mechanism of interfacial superconductivity.

According to the electrostatic model and cooperative mechanism[21,29,30], it is proposed that interfacial superconductivity is related with the the electron correlation effect of the superconducting layer, the dielectric constant of the substrate, and the interlayer distance between them. Electron correlation is prerequisite for $T_g$ enhancement[28,30], which has been demonstrated in FeS/STO showing weak electron correlation and absence of superconductivity[52]. Based on the similar bandwidth observed in FeSe/LFO and FeSe/STO (Fig. 2), the electron correlation effect is similar. At this correlation strength, the superconductivity is weakly dependent on the dielectric constant of the substrate in the range of 30–10,000 (Rosenstein and Shapiro[30]), and the dielectric constant of LFO locates within this range[53,54]. Therefore, the critical difference between FeSe/LFO and FeSe/STO is the interlayer distance between FeSe and the adjacent oxide. The smaller interlayer distance in FeSe/LFO (2.5 Å) than that in FeSe/STO (2.9 Å)(Peng et al.[37]) indicates stronger interfacial bonding[55] and would enhance interfacial EPC and superconductivity according to theories[21,29,30], which are consistent with our experimental observations. Note that 2.5 Å is still quite large compared with the inter-atomic distances of regular chemical bonds (Fig. 1e), and there are still rooms for further increasing the interfacial bonding strength and enhancing interfacial EPC. This points out the direction for further enhancing superconductivity through heterostructure designing.

Although the interfacial EPC could adequately explain the significant enhancement of pairing in FeSe/LFO, the role played by the spin fluctuations in the LFO substrate might not be excluded yet. Note that LFO bulk is a Mott insulator with G-type antiferromagnetism and remains antiferromagnetic when grown as a film on STO(100) with thickness down to 2–4 nm (refs. [56,57]). It was theoretically proposed that at the interface where antiferromagnetic order diminishes, the spin fluctuation is strong, which may further enhance superconductivity[35,58]. As RIXS is capable of measuring the spin excitations of 1uc FeSe/STO (Pelliciari et al.[34]), our work calls for future RIXS studies on the spin excitation of FeSe/LFO interface, in order to understand whether the spin excitation of LFO takes extra effects on superconductivity of the FeSe film.

To summarize, combining comprehensive sets of experimental measurements on the same samples, high-temperature superconductivity is shown to present in single-layer FeSe/LFO/STO with sharp FeSe-$FeO_x$ interface. The large Cooper pairing strength and the diamagnetic response demonstrate that FeSe/LFO interface shows an enhanced superconducting phase as compared to FeSe/STO. Superconductivity is delicate and inhomogeneous, but the highest-achievable superconducting pairing potentially persists above liquid nitrogen boiling temperature. Though the insulating nature of LFO poses challenges on ARPES studies, such a superconductor-insulator heterostructure is however a merit for future designing of cost-effective superconducting device. Moreover, our data show that the cooperative pairing enhancement mechanism can work well beyond FeSe-$TiO_x$, and the larger enhancement of superconducting pairing can be attributed to the stronger interfacial EPC in FeSe/LFO. The strong and selective interaction between FeSe electrons and LFO phonons with shorter interfacial bond length points the direction for optimizing superconductivity through heterostructure designing.

## Methods

**Sample preparation.** Commercial 0.5%wt Nb doped $SrTiO_3$ (STO) substrates were etched and annealed to get $TiO_x$ terminated surfaces. Then 6uc of LFO films were epitaxially grown on STO by PLD. The film thickness of LFO was confirmed by real-time intensity oscillation of reflective high-energy electron diffraction (RHEED) and x-ray reflectivity studies (see Supplementary Note 1). In order to ground the FeSe layer for ARPES study, gold film was sputtered at the LFO/STO edge and then covered with silver paste. After that, the LFO/STO substrate was transferred to the MBE chamber with a base pressure of $7 \times 10^{-10}$ mbar. The substrate was degassed at 550 °C and heated to 950 °C under Se flux for 45 min. Then 1uc FeSe film was grown on LFO/STO at 490 °C by co-evaporation method with a Fe/Se flux rate ratio of 1:10 and annealed at 520 °C for 3 h. After ARPES measurements, the sample was measured by in-situ STM and then transferred back to the MBE chamber for growth of an additional amount of 0.5uc FeSe, and the 1.5uc FeSe/LFO/STO was measured by in-situ ARPES and STM again. Thereafter, the sample was capped with Se at room temperature and taken out from the vacuum chamber, and cut into several pieces for ex-situ mutual inductance measurements, cross-sectional STEM studies, and reflected HREELS measurements. The complete data of the same sample involving ARPES, STM, STEM, mutual inductance, and HEELS experimental techniques ensure the credibility and stability of the analysis. The ARPES experiments were repeated on different samples under similar growth and annealing procedures, and ARPES data on another 1uc FeSe/6uc LFO/STO samples is shown in Supplementary Fig. 6.

**ARPES measurements.** In-situ ARPES measurements were performed on FeSe/LFO/STO with a Fermi Instruments 21.2 eV helium discharge lamp, using a Scienta DA30 electron analyzer, under a typical vacuum of $2.5 \times 10^{-11}$ mbar. The total energy resolution is 6 meV and angular resolution is 0.3°.

ARPES data on another 1uc FeSe/6uc LFO/STO samples is shown on Supplementary Fig. 6, which was grown at the same condition as described above in a different MBE system, and measured by in-situ ARPES with a Fermi Instruments 21.2 eV helium discharge lamp, using a R4000 electron analyzer, under a typical vacuum of $4 \times 10^{-11}$ mbar.

**STEM measurements.** TEM sample was prepared by using Focused Ion Beam (FIB) milling. Cross-sectional lamella was thinned down to 100 nm thick at an accelerating voltage of 30 kV with a decreasing current from the maximum 2.5 nA, followed by fine polish at an accelerating voltage of 2 kV with a small current of 40 pA. The atomic structures of the FeSe/LFO/STO films were characterized using an ARM–200CF (JEOL, Tokyo, Japan) transmission electron microscope operated at 200 kV and equipped with double spherical aberration (Cs) correctors. HAADF images were acquired at acceptance angles of 90–370 mrad.

**Reflected HREELS measurements.** The Se-capped FeSe/LFO/STO samples were transferred to a HREELS system[59], and annealed at 450 °C for 4 h to remove the Se capping layer. Low-energy electron diffraction (LEED) patterns were collected to confirm the removal of the capping layer and verify the sample quality. HREELS measurements, with an energy resolution of 3 meV, were performed at 35 K, with an incident beam energy of 110 eV and an incident angle of 60° with respect to the surface normal.

**Mutual inductance measurements.** The sample was sandwiched between two concentric coils for detecting the diamagnetic signal. The AC current applied to the drive coil has an amplitude of 1 μA and a frequency of 33.1 kHz. The temperature dependent mutual inductance measurements were conducted from 2 to 120 K with the temperature increasing rate 1 K/min.

**Scanning Tunneling measurements.** The topography measurements were performed in situ using a STM (RHK Technology) that connected with the ARPES chamber and MBE chambers in a combined ultra-high vacuum (UHV) system. The samples were measured at 17 K under a vacuum of ~$1 \times 10^{-10}$ mbar. Scanning tunneling spectroscopy (STS) was not performed due to the high noise ratio of the STM system connected to the ARPES-MBE system.

## Data availability

Relevant data supporting the key findings of this study are available within the article and the Supplementary Information file. All raw data generated during the current study are available from the corresponding authors on reasonable request.

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

## Acknowledgements

We gratefully acknowledge the experimental support by M.Y.N. Lei, D.M. Zhao, Q.Q. Ge and W.W. Sun. This work is supported in part by the National Natural Science Foundation of China (Grants No. 11790310, No. 11922403, No. 11888101, No. 12074074, No. 52025025, and No. 11634016), the National Key R&D Program of the MOST of China (Grants No. 2017YFA0303004 and No. 2016YFA0300200), Shanghai Rising-Star Program (Grant No. 20QA1401400), Shanghai Municipal Science and Technology Major Project (Grant No. 2019SHZDZX01), Beijing Natural Science Foundation (Grant No. Z190010), and Anhui Initiative in Quantum Information Technologies.

## Author contributions

Y.S., X.C., X.L., and T.Y. grew the FeSe films. Z.C. and Y.X. grew the LFO/STO heterostructure. Q.Z. and L.G. performed STEM measurements. X.X., X.Z., and J.G. measured the HREELS. H.R. and Y.W. did the mutual inductance experiments. Y.S., R.T., and T.Z. characterized the samples by STM. Y.S., X.L., and R.P. performed ARPES measurements. Y.S., H.X., R.P., and D.F. analyzed the ARPES data. Y.S., R.P., and D.F. wrote the paper. R.P. and D.F. are responsible for the infrastructure, project direction and planning. All authors have discussed the results and the interpretation.

## Competing interests

The authors declare no competing interests.
