## [Peer Review File · Nature Communications]

REVIEWER COMMENTS

Reviewer #1 (Remarks to the Author):

In their manuscript, Song et al claim that they observe superconductivity in 1 uc of FeSe grown on LaFeO₃ with a T_c as high as 80 K. Their main claim is based on ARPES and diamagnetic response data. Discovery of 80K superconductivity on a new substrate is an important finding in the study of 1 ML superconductivity as it may show the way to achieve even higher T_c superconductivity (which is discussed in the manuscript). The data presented in the manuscript, taken using various techniques, are of very high quality, probably a result of careful and time-consuming work. One has to make it sure that the observations are results of superconductivity, not artifacts. For that matter, I would say that Figure 3k is the key figure that shows the observations are results of superconductivity. I find this work quite interesting and would recommend for publication.

With my recommendation stated, I have comments/questions for the authors to consider to improve the manuscript.

1. In figure 3k, authors compare 1uc FeSe/STO with 1.5uc FeSe/LFO. Why not 1uc FeSe/LFO? Since the measurement does not suffer from charging, it would be better to present 1uc FeSe/LFO; it not only is a fair comparison but also should show higher T_c.
2. The authors do not seem to discuss strain effect from the substrate. How do lattice parameters of SrTiO₃ and LaFeO₃ compare? Any strain from the substrate can affect the alpha angle which is an additional (in addition to the doping and EPC discussed in the manuscript) parameter which affects the superconductivity.
3. The discussion given in the paragraph that starts with "To reconcile the observation of T_g~51K...." is a bit premature. If SC gap in ARPES is not seen because the SC portion is small as the authors argue, then I would say that the ARPES gap should slowly increase. However, the gap suddenly develops at 51 K as figure 3h shows. What is special about 51 K (so that gap suddenly starts to develop)? I would recommend moving this discussion to the discussion section as it is not really clear. It is usually a good idea to separate the experimental results and their interpretation.
4. The authors seem to suggest shorter bonding distance should result in a stronger EPC. But for a higher EPC, what we need is a stronger bonding, not shorter bonding length. Is the bonding between FeSe and LFO stronger than that between FeSe and STO?
5. Authors show STM images in figures 3a and 3d. Have they performed STS? Is it more prone to the charging problem? Authors could mention this in the manuscript.
6. If I understood correctly. authors suggest that the work function difference between FeSe (5.1 eV) and STO (4.5 eV) (or LFO (4.6 eV)) derive the electrons to FeSe. However, the work function value of an insulator depends on the Fermi level position within the gap. Are authors sure that the work functions of their substrates are also 4.5 and 4.6 eV? Have they measured?
7. Previous studies on 1ML FeSe grown on various substrates (MgO and NGO) are discussed in the manuscript. Since the authors correlate the work function and charge transfer, it would be important to check out the failed cases. That is, what are the work functions for MgO and NGO? Are those values compatible with the scenario that the work function difference determines the amount of charge?

Reviewer #2 (Remarks to the Author):

The manuscript by Song et al., titled "High-temperature superconductivity at FeSe/LaFeO₃ interface," reports the successful growth of FeSe monolayers on a 6 UC LaFeO₃ (LFO) film [which is in turn grown on SrTiO₃ (STO)], and its subsequent characterization. By imaging the replica bands and monitoring the superconducting gap, the authors claim that this system achieves a higher T_c compared to FeSe/STO and that this enhancement is due to stronger electron-phonon coupling between the FeSe

electrons and the substrate phonons.

At this time, I cannot make a definitive recommendation as I have several questions and comments that the authors need to address. Before I provide them, I would like to make a general point. To date, enhanced T_c 's in FeSe monolayers have only been achieved using TiO-based substrates like BaTiO₃, TiO₂, and SrTiO₃, and the lack of additional examples has limited our ability to discriminate between competing proposals for the T_c enhancement. The system discussed in this work, which nominally terminates at a FeO layer, can provide an excellent point of comparison. The current study seems to realize this but doesn't do a good enough job discussing what I think are inconsistencies with the current cross-interfacial electron-phonon coupling model. Many of my comments will be along these lines.

1) As far as I can tell, only samples involving 6 UC of LFO have been presented. Why is this? Are these results robust against thicker or thinner LFO layers? Have those control experiments been done?

2) I find the width of the 94 meV Fuchs-Kliwer (FK) mode shown in figure 4d perplexing, and possibly speaks against the author's interpretation of the data. The large broadening of the FK mode is attributed to the coupling between the STO lattice and the FeSe electrons. However, the results shown in Fig 4d suggest that the width of the STO FK mode is the same for both FeSe/STO and FeSe/LFO/STO samples. This suggests that the coupling to the FK mode is unchanged, despite the fact that you have inserted six UC between the STO and FeSe layers and increased the distance between the STO phonons and the FeSe electrons. The authors attribute this to a lack of screening by the LFO layer. The observed behavior therefore seems inconsistent with the author's interpretation that placing FeSe closer to the substrate should increase the coupling to those modes. Do the authors have any idea why this is? Can they comment? Also, I would expect the oxygen orbitals in LFO to provide some degree of screening due to their polarizable nature. Why is that not active here?

3) The discussion and interpretation of the e-ph coupling need to be re-considered in many places. First of all, there is a typo in the expression for $g(q)$ appearing in Ref. 5 (which was propagated to Ref 19), as pointed out by Kubic & Dolgov in NJP 19, 013020 (2017). The latter reference should be cited along with the others when the authors discuss the functional form of $g(q)$ to that a reader can find it. Second, the discussion of the results equates a closer distance to the substrate with stronger e-ph coupling. This equivocation is incorrect and very misleading. Assuming the electrostatic coupling proposed in Ref 5 is correct, then the total coupling strength has an optimal value at some distance and depends strongly on the value of the substrate's dielectric constant. These issues are discussed nicely in a recent preprint by Rosenstein and Ya Shapiro (arXiv:2102.03854). To conclude anything about the strength of the e-p coupling, one must have some idea of the dielectric function value. Alternatively, one could try to examine the spectral weight of the replica bands, as discussed in Ref. 21.

For example, are the spectral weights of the replica bands consistent with the values inferred by examining the band splitting on page 7?

4) Related to the above point, the authors write about the cross-interfacial e-ph coupling scenario as if it's a given. However, it remains quite controversial. For example, work from George Sawatzky's group suggests that the replica bands may have nothing to do with e-ph coupling [PRL 120, 237001], while recent work from Kyle Shen's group [arXiv:2102.03397] seems to speak against this proposal. Similarly, the authors should mention the recent FLEX works [PRB 102, 180501(R); PRB 103, 144504], which arrive at quite different conclusions on whether forward scattering can be combined with spin fluctuations in this system to enhance T_c . These works show that the cross-interfacial e-p coupling scenario is far from accepted, and the reader should, at the very least, be made aware of these issues. Later in the paper, the authors also elude performing RIXS measurements to study the spin

excitations; these measurements have in fact, been done, see Pelliciani et al., arXiv:2008.09618.

5) It would be valuable to the community to compare the current results against the reference FeSe/STO interface (which superconducts) and FeS/STO interface, which does not [see K. Shigekawa et al, PNAS 116, 2470 (2019)]. These two systems also have different distances between the FeSe layer and the STO substrate. How do these compare?

5) Throughout the paper, the authors refer to Ref. 26 as having performed "approximation free quantum Monte Carlo simulations." I strongly object to this description. When calculating the T_c enhancements, the authors of that paper introduced an electronic pairing interaction by hand. Moreover, they considered finite-size lattices, which approximate the thermodynamic limit. Both of these are approximations. I suggest that the authors remove instances of "approximation free" throughout the paper and instead refer to them as QMC calculations.

Referee report on “High temperature superconductivity at FeSe/LaFeO₃ interface“

In this manuscript, the authors engineered a new FeSe/LaFeO₃ interface to study the role of interfacial electron-phonon coupling (EPC) in superconductivity enhancement. Angle-resolved photoemission spectroscopy and mutual inductance measurements reveal superconducting pairing at a higher temperature (~80 K) than that which was previously observed in FeSe/SrTiO₃ (~65 K). The authors suggests that the shortened interfacial bond length leads to stronger EPC, and this result may be used in future studies to engineer new interfaces that enhance superconductivity.

I find that the paper is well-motivated and the researchers have completed a thorough set of measurements, which leads to a reasonably complete and coherent story about superconductivity at the FeSe/LaFeO₃ interface.

However, I believe the results could be explained more clearly. Specifically, in the part about the band splitting in the ARPES data shown in Fig. 2. It would also be important to expand on it and clarify the conclusion regarding superconductivity enhancement through the heterostructure design.

Some other comments and questions:

1. In Figure 1, what are some of the theoretical explanations for the origin of the band splitting at the Γ and M points? Presumably it is related to the electron-phonon coupling, which would be an interesting finding - therefore, it is worthwhile to explain this feature more thoroughly in the text.
2. Furthermore, it is unclear why this splitting is an indication of a high-quality sample. E.g. could such splitting be an indicator that the analyzer has sufficient energy and momentum resolution.
3. Have there been any DFT calculations done on this structure? If so, do the calculations predict the observed band splitting?
4. It would be good to see the RHEED pattern for the 1.5 μ c sample as a point of comparison to the 1 μ c sample.
5. Is it possible to improve the interconnection of the FeSe films by changing the growth condition e.g. growing at a higher temperature or laser power? Some discussion would be good to see.

Once these issues listed above are resolved I can recommend this manuscript for publication.

REVIEWER COMMENTS

Reviewer #1 (Remarks to the Author):

In their manuscript, Song et al claim that they observe superconductivity in 1 uc of FeSe grown on LaFeO₃ with a T_c as high as 80 K. Their main claim is based on ARPES and diamagnetic response data. Discovery of 80K superconductivity on a new substrate is an important finding in the study of 1 ML superconductivity as it may show the way to achieve even higher T_c superconductivity (which is discussed in the manuscript). The data presented in the manuscript, taken using various techniques, are of very high quality, probably a result of careful and time-consuming work. One has to make it sure that the observations are results of superconductivity, not artifacts. For that matter, I would say that Figure 3k is the key figure that shows the observations are results of superconductivity. I find this work quite interesting and would recommend for publication.

Reply: We thank the reviewer for summarizing our results and pointing out that our results are of very high quality and obtained by careful and time-consuming work. We appreciate the reviewer's recommendation.

With my recommendation stated, I have comments/questions for the authors to consider to improve the manuscript.

1. In figure 3k, authors compare 1uc FeSe/STO with 1.5uc FeSe/LFO. Why not 1uc FeSe/LFO? Since the measurement does not suffer from charging, it would be better to present 1uc FeSe/LFO; it not only is a fair comparison but also should show higher T_c.

Reply: We thank the Reviewer for pointing this out. However, as the mutual inductance measurements were conducted ex-situ, capping layers of both Se and additional FeSe layers are needed, in order to protect the film from degrading at atmosphere and without destroying the superconductivity of the 1uc film. Using Se as capping layer, superconductivity has only been reported when FeSe is more than 1uc thick [Z. Zhang, et al., Sci. Bull. 60, 1301–1304 (2015)]. As a comparison, the FeSe/STO sample in Figure 3k is also not 1uc-FeSe/STO, but has a complex protection layer that consists of 18nm-Se/2uc-FeSe/2uc-(Fe_{0.96}Co_{0.04})Se [Z. Zhang, et al., Sci. Bull. 60, 1301–1304 (2015)].

Following the reviewer's suggestion, we conducted mutual inductance measurement on a 1uc-FeSe/LFO sample (the superconducting gap of the same sample is shown in Supplementary Sec. 4) with Se capping, but failed to obtain any superconducting signal (Fig. R1). The measured signal is similar to the background signal because the superconductivity of 1uc FeSe film is easily destroyed by directly contacting with the Se capping [Y. T. Cui, et al., PRL 114, 037002 (2015)]. While in the 1.5uc FeSe sample, the 2nd uc FeSe islands can protect the stoichiometry of 1st uc FeSe layer (see the STEM-HAADF results in Fig. 1 in the main text, where the well-ordered FeSe layer is protected beneath the layer of a mixture of 2nd uc

FeSe and amorphous Se), so it can still provide diamagnetic signal during mutual inductance measurement.

Fig. R1 Temperature dependence of the out-of-phase voltage V_{out} in the pickup coil measured by the ex-situ mutual inductance experiments on the amorphous-Se capped 1uc FeSe/LFO/STO, amorphous-Se capped 1.5uc FeSe/LFO/STO, and the comparative sample of Se capped LFO/STO with the same heat treatment.

In the revised manuscript, we have clarified that the composition of the comparative FeSe/STO sample is Se/2uc-FeSe/2uc-Fe_{0.96}Co_{0.04}Se/1uc-FeSe/STO in Fig. 3k. Besides, we have stated that “the additional 0.5uc FeSe layer protects the underneath 1uc FeSe from directly contacting with the Se capping and preserve its stoichiometry (Fig. 1), which is crucial for ex-situ superconductivity measurements”.

2. The authors do not seem to discuss strain effect from the substrate. How do lattice parameters of SrTiO₃ and LaFeO₃ compare? Any strain from the substrate can affect the alpha angle which is an additional (in addition to the doping and EPC discussed in the manuscript) parameter which affects the superconductivity.

Reply: We thank the reviewer for this highly relevant question. The lattice parameters are $a=3.905\text{\AA}$ for STO and $a=3.93\text{\AA}$ (pseudo cubic lattice) for bulk LFO [A. Scholl, et al., Science 287, 1014 (2000)]. With such small lattice mismatch of 0.64%, epitaxial 6uc LFO film is expected to be strained to the STO lattice. Consistently, from large scale STEM image (Fig. R2), both the in-plane lattices of FeSe and 6uc LFO well match that of the STO substrate, which suggests both FeSe and LFO are coherently strained to the STO lattice (Fig. R2). The anion height and the in-plane lattice are both similar to those in FeSe/STO [R. Peng, et al., Sci. Adv. 6, eaay4517 (2020)], therefore the alpha angle of FeSe layer remain similar with respect to FeSe/STO.

Following the Reviewer’s suggestion, we have added the above discussions into the revised manuscript and the large-scale STEM image in our revised Supplementary Information.

Fig. R2 STEM-HAADF image of the Se-capped FeSe/LFO/STO sample in a larger scale shows no visible relaxation in the LFO layer.

3. The discussion given in the paragraph that starts with “To reconcile the observation of $T_g \sim 51\text{K} \dots$ ” is a bit premature. If SC gap in ARPES is not seen because the SC portion is small as the authors argue, then I would say that the ARPES gap should slowly increase. However, the gap suddenly develops at 51 K as figure 3h shows. What is special about 51 K (so that gap suddenly starts to develop)? I would recommend moving this discussion to the discussion section as it is not really clear. It is usually a good idea to separate the experimental results and their interpretation.

Reply: We thank the reviewer for pointing out the unclear part of the discussion on the sample inhomogeneity.

First, we agree with the reviewer that the paragraph is an interpretation rather than the experimental results, which should be moved to the discussion part.

Second, we would like to clarify why the “ARPES gap” does not seem to “slowly increase”. As mutual inductance results show a dramatic drop also around 51K, most regions of the sample could form Cooper pairing at 51K, which corresponds to gap opening at 51K. Ideally there should be a partial gap opening between 51-80K if there is a small portion in the sample possessing $T_g \sim 80\text{K}$; however, “the slowly increasing gap” between 51K and 80K could only be directly observed when the superconducting peak width is sufficiently small. In experiments with significant thermal broadening and large normal state contribution, the small portion of gap opening would make the symmetrized EDC broader with a flatter peak top, rather than a gap, which is shown in the following simulation. Supposing all regions of the sample homogeneously showing $T_g \sim 51\text{K}$, we simulate the symmetrized EDC of 61K with the parameters obtained by fitting the 31K data, but by setting $\text{Gap} = 0\text{meV}$ and a temperature broadening to 61K. The measured data is broader than the simulated curve, indicating an additional broadening in the 61K data possibly related with a partial gap opening. In Fig. R3(e), supposing 1/4 of the sample already opens a gap $\sim 17\text{meV}$ at 61K, we simulate the gap

function with Gap=17meV, and adding this spectral function to the Gap=0meV simulated curve with an intensity ratio of 1:3, we can see that the total intensity matches the measured data quite well.

Although the simulation can well demonstrate the partial gap opening, it is not a rigorous demonstration of the ratio of the Tg~80K regions, because the real case is more complicated than the simulation. For example, the contribution from the superconducting spectral function with $\Delta=17\text{meV}$ is not considered in the fitting of the 31K data. Besides, while estimating the portion of superconducting regions, the whole sample should be considered; however, the 1uc FeSe portion covered by the second unit cell FeSe in the sample contributes little to ARPES data and cannot be simulated here. Due to the portion of the high Tg region is small, it is reasonable that the slowly increment of gap at 80K is not observed in the data, as shown by our simulation. We thank the reviewer's comments which make our discussion more rigid.

Fig. R3 Simulation of partial gap opening below 80K due to a small portion of sample with Tg~80K.

Following the reviewer's suggestion, we have moved the interpretation to the discussion section in the revised manuscript. Besides, we have added "A partial gap opening is expected between 51 K and 80 K if there is a small portion in the sample possessing Tg~80K, but it is hard to observe due to the much higher intensity of normal state spectra" in the revised manuscript, and included the Fig. R3 and the above detailed discussions in the revised Supplementary Information.

4. The authors seem to suggest shorter bonding distance should result in a stronger EPC. But for a higher EPC, what we need is a stronger bonding, not shorter bonding length. Is the

bonding between FeSe and LFO stronger than that between FeSe and STO?

Reply: We thank the reviewer for pointing this out. Considering the bonding between two atoms, generally, as the bond strength increases, the bond length decreases [P. Flowers, Paul, et al., *Chemistry: Atoms First 2e*, (2015)]. In the case of FeSe/STO, first-principle calculations suggest the interlayer distance become shorter with stronger interfacial bonding [J. Bang, et al., *PRB* 87(22), 220503 (2013)]. As Fe shows similar ionic radius as Ti, the starting distance before bonding between FeSe and FeOx should be similar to that of FeSe and TiOx, and shorter bonding distance generally reflect stronger bonding.

Moreover, interlayer distance and EPC strength can be directly related according to theoretical calculations. According to the estimation of the interfacial EPC strength from theoretical papers [J. J. Lee, et al., *Nature* 515, 245–248 (2014), D.-H. Lee, *Chin. Phys. B* 24, 117405 (2015), Kulic & Dolgov, *NJP* 19, 013020 (2017), Baruch Rosenstein, et al., *PRB* 103, 224517 (2021)], under the circumstance that other conditions are the same, the smaller interlayer distance h corresponds to the stronger EPC. Baruch Rosenstein, et al. [*PRB* 103, 224517 (2021)] also pointed out this (“In view of the exponential SCP, Eq. (1), the EPI pairing in FeS/STO is weaker than in FeSe/STO since the distance between 2DEG and the TiO₂ layer increases [13] by 15%. This alone should reduce the EPI coupling.”).

Following the reviewer’s suggestion, we have cited the above references and added “the smaller interlayer distance in FeSe/LFO (2.5Å) than that in FeSe/STO (2.9Å) indicates stronger interfacial bonding and would enhance interfacial EPC and superconductivity according to theories” in the revised manuscript.

5. Authors show STM images in figures 3a and 3d. Have they performed STS? Is it more prone to the charging problem? Authors could mention this in the manuscript.

Reply: The RHK-STM that we used is connected to our combined ARPES-OMBE-Chalcogenide MBE system, so it is not good at vibration isolation and has strong noise during STS measurements. The experimental condition is just enough for scanning the surface topography but challenging for STS measurements. We did not measure STS on the sample we presented in our manuscript, and we are not certain whether it is more prone to the charging problem or not. We have added following text to the revised manuscript: “STS was not performed due to the high noise ratio of the STM system connected with ARPES-MBE system”.

6. If I understood correctly, authors suggest that the work function difference between FeSe (5.1 eV) and STO (4.5 eV) (or LFO (4.6 eV)) derive the electrons to FeSe. However, the work function value of an insulator depends on the Fermi level position within the gap. Are authors sure that the work functions of their substrates are also 4.5 and 4.6 eV? Have they measured?

Reply: We thank the Reviewer for pointing this out. We are not able to measure the work function in our ARPES system. In the setup of our ARPES system (and most other ARPES),

both sample and analyzer are grounded, and thus the measured work function corresponds to that of the analyzer rather than that of the samples [J. A. Sobota, et al., *Rev. Mod. Phys.* 93, 025006(2021)]. To measure the work function of a sample by photoemission requires floating of the sample and applying voltage between the sample and analyzer, which was demonstrated in a previous report in measuring the work function of FeSe and STO [H. Zhang, et al., *Nat. Commun.* 8, 214 (2017)] but cannot be realized in our current experimental setup.

The work function of LFO is from previous literature of its bulk material and determined by combining XES and XPS [W. T. Hong, et al., *J. Phys. Chem. C*, 119, 2063–2072 (2015)]. The work functions of STO and FeSe are from the UPS results of STO and thick FeSe film on STO [H. Zhang, et al., *Nat. Commun.* 8, 214 (2017)].

That the work function difference between FeSe and STO drives the electrons to FeSe is a viewpoint proposed by previous FeSe/STO studies [H. Zhang, et al., *Nat. Commun.* 8, 214 (2017), W. Zhao, et al., *Sci. Adv.* 4, eaao2682 (2018)]. We also observed the energy shift in EELS result of FeSe/LFO. Combining them with the reported work function of LFO bulk material and the electron doping level derived from ARPES, these results in the three aspects suggest that the pictures of interfacial charge transfer in FeSe/LFO and FeSe/STO are qualitatively consistent. We agree with the reviewer that precisely determine the work function value of FeSe and LFO experimentally would be crucial in checking the scenario quantitatively.

In the revised manuscript, we have clearly stated that the work function values of LFO and FeSe are from previous report on bulk materials or thick films. Following the suggestion of the Reviewer, we have also added “these results suggest that the interfacial charge transfer in FeSe/LFO can be qualitatively accounted by the work function mismatch scenario similar to FeSe/STO, while a quantitative comparison call for a precise determination on the work function of each ingredient in the heterostructure”.

7. Previous studies on 1ML FeSe grown on various substrates (MgO and NGO) are discussed in the manuscript. Since the authors correlate the work function and charge transfer, it would be important to check out the failed cases. That is, what are the work functions for MgO and NGO? Are those values compatible with the scenario that the work function difference determines the amount of charge?

Reply: The reported work function for MgO(001) is 4.94 eV [J. Y. Lim et al., *J. Appl. Phys.* 94, 764 (2003)]. It is closer to the FeSe work function 5.1 eV than the LFO work function 4.6eV, which could lead to a lower level of electron doping in FeSe/MgO. We have not found the reported work function for NGO in previous literatures, so it is not clear about the work function difference in FeSe/NGO. On the other hand, no ARPES measurements have ever been done in the FeSe/MgO and FeSe/NGO samples and the actual amount of charge that was doped into FeSe is unknown, so by the work function values alone we cannot certainly determine whether the work function difference is compatible with the amount of

charge in these systems.

Reviewer #2 (Remarks to the Author):

The manuscript by Song et al., titled "High-temperature superconductivity at FeSe/LaFeO₃ interface," reports the successful growth of FeSe monolayers on a 6 UC LaFeO₃ (LFO) film [which is in turn grown on SrTiO₃ (STO)], and its subsequent characterization. By imaging the replica bands and monitoring the superconducting gap, the authors claim that this system achieves a higher T_c compared to FeSe/STO and that this enhancement is due to stronger electron-phonon coupling between the FeSe electrons and the substrate phonons.

At this time, I cannot make a definitive recommendation as I have several questions and comments that the authors need to address. Before I provide them, I would like to make a general point. To date, enhanced T_c's in FeSe monolayers have only been achieved using TiO-based substrates like BaTiO₃, TiO₂, and SrTiO₃, and the lack of additional examples has limited our ability to discriminate between competing proposals for the T_c enhancement. The system discussed in this work, which nominally terminates at a FeO layer, can provide an excellent point of comparison. The current study seems to realize this but doesn't do a good enough job discussing what I think are inconsistencies with the current cross-interfacial electron-phonon coupling model. Many of my comments will be along these lines.

Reply: We appreciate the reviewer for pointing out that our system provides an excellent point of comparison. We also thank the reviewer for his/her suggestions on the improvement of the discussion part of our manuscript. According to the reviewer's following suggestions/comments, we have modified the discussion about the cross-interfacial electron-phonon coupling model and made it more rigorous in the revised manuscript.

1) As far as I can tell, only samples involving 6 UC of LFO have been presented. Why is this? Are these results robust against thicker or thinner LFO layers? Have those control experiments been done?

Reply: We thank the Reviewer for pointing this out. We intended to study the single layer FeSe films that are epitaxial on FeO_x interfaces instead of TiO_x. From the cross-sectional STEM results (Fig. 1 in main text), we found there is a 2uc intermixing between the Fe and Ti around the interface. Therefore, if the LFO is less than 2uc there will be a chance that some TiO_x surface was exposed to FeSe, which is unfavorable for focusing on the study of pure FeSe/FeO_x interface. In our experiments, the thickness of LFO we used is 6uc and thicker to avoid possible exposure of TiO_x.

The band structure observed in single layer FeSe grown on thicker LFO is generally robust. On a substrate of 12uc LFO/STO, we also obtained single layer FeSe film with good

quality (Fig. R4). Its band structure is similar to the 1uc FeSe grown on 6uc-LFO/STO. However, a thicker insulating LFO layer caused severer charging problem during our ARPES measurement, which resulted in a large charging-induced energy shift and some distortion of the bands (Fig. R4). Under the full light intensity, it showed a charging shift at the level of hundreds meV, which obstructed the superconducting gap analysis. We tried several pieces of 12uc LFO but still cannot overcome the charging problem in them. LFO thickness dependent effect on the superconductivity of 1uc FeSe film deserves further study, but is not our main focus of the work in this manuscript.

Following the Reviewer's suggestion, in the revised manuscript, we have explained the reason why we chose 6uc-LFO/STO as substrates for growing single layer FeSe films.

Fig. R4 Band structure at Gamma (a) and M (b) points of an 1uc FeSe grown 12uc-LFO/STO sample measured by ARPES (T=30K).

2) I find the width of the 94 meV Fuchs-Kliwer (FK) mode shown in figure 4d perplexing, and possibly speaks against the author's interpretation of the data. The large broadening of the FK mode is attributed to the coupling between the STO lattice and the FeSe electrons. However, the results shown in Fig 4d suggest that the width of the STO FK mode is the same for both FeSe/STO and FeSe/LFO/STO samples. This suggests that the coupling to the FK mode is unchanged, despite the fact that you have inserted six UC between the STO and FeSe layers and increased the distance between the STO phonons and the FeSe electrons. The authors attribute this to a lack of screening by the LFO layer. The observed behavior therefore seems inconsistent with the author's interpretation that placing FeSe closer to the substrate should increase the coupling to those modes. Do the authors have any idea why this is? Can they comment? Also, I would expect the oxygen orbitals in LFO to provide some degree of screening due to their polarizable nature. Why is that not active here?

Reply: We thank the reviewer for raising these important questions.

First, we agree with the reviewer that the oxygen orbitals in LFO should provide some degree of screening due to their polarizable nature. Indeed, experimentally, with the total intensity normalized by the elastic peak, the peak height of the 94-95meV phonon in

FeSe/LFO/STO is 0.22 to that of FeSe/STO (note that the red line is scaled by 0.22 in Fig. 4d). The decreased peak height is due to the partially screened FK field of STO by LFO, which leads to the decay of the field intensity [S. Y. Zhang, et al., Phys. Rev. B 94, 081116 (2016)].

Second, as for “width of the STO FK mode is the same for both FeSe/STO and FeSe/LFO/STO samples”, we do not think this can indicate a similar electron-phonon interaction between FeSe and STO in these two different samples. The phonon lifetimes of STO and LFO/STO samples are different to start with. As shown in Fig. 7 of S. Y. Zhang, et al., Phys. Rev. B 97, 035408 (2018), without grown FeSe, different oxides already show dramatically different FWHM of the phonon peaks. Moreover, although the EPC would affect the width of the FK mode, its contribution to the peak width is rather small, while the peak width is largely affected by other factors such as sample quality. As mentioned in S. Y. Zhang et al., Phys. Rev. B 97, 035408 (2018), the linewidth of FeSe/STO varies with the substrate batches and growth conditions. Below in Fig. R5, we show the comparison between two 1uc FeSe/STO samples and our 1uc FeSe/LFO/STO. The peak width of different FeSe/STO already varies a lot. Compared with the blue curve, the FeSe/LFO/STO sample show a much-reduced peak width of the STO mode, while is similar to the red curve. Therefore, sample-dependent variation on the peak width is much larger than the effect from EPC strength. Since we have not measured the peak width of the same sample before FeSe growth, and we think it is unreliable to claim the EPC strength by comparing peak width of different samples.

Fig. R5 Comparison of the peak width between different FeSe/STO and FeSe/LFO/STO samples. The curves are normalized by the peak height of the α mode of the STO FK phonon.

In the revised manuscript, to make it more obvious, we have stated the caption that the EELS data of FeSe/STO is multiplied by a factor of 0.22, and claimed the partial screening of STO phonons in the main text.

3) The discussion and interpretation of the e-ph coupling need to be re-considered in many places. First of all, there is a typo in the expression for $g(q)$ appearing in Ref. 5 (which was propagated to Ref 19), as pointed out by Kulic & Dolgov in NJP 19, 013020 (2017). The latter reference should be cited along with the others when the authors discuss the functional form of $g(q)$ to that a reader can find it. Second, the discussion of the results equates a closer distance to the substrate with stronger e-ph coupling. This equivocation is incorrect and very

misleading. Assuming the electrostatic coupling proposed in Ref 5 is correct, then the total coupling strength has an optimal value at some distance and depends strongly on the value of the substrate's dielectric constant. These issues are discussed nicely in a recent preprint by Rosenstein and Ya Shapiro (arXiv:2102.03854). To conclude anything about the strength of the e-p coupling, one must have some idea of the dielectric function value. Alternatively, one could try to examine the spectral weight of the replica bands, as discussed in Ref. 21.

For example, are the spectral weights of the replica bands consistent with the values inferred by examining the band splitting on page 7?

Reply: We thank the Reviewer for pointing out the related references and constructive comments. According to the references [Ref. 5, Ref. 19, Kulic & Dolgov in NJP 19, 013020 (2017) and Rosenstein and Ya Shapiro, PRB 103, 224517 (2021)], $g(q)$ increases with the decreasing interlayer distance h when electron correlation and dielectric constant are in a proper range. This conclusion is explicitly stated in the recent work by Rosenstein and Ya Shapiro as pointed out by the Reviewer [Rosenstein and Ya Shapiro, PRB 103, 224517 (2021)]. Figure 2 in that paper [Rosenstein and Ya Shapiro, PRB 103, 224517 (2021)] also presented the relation between T_c and h .

We also thank the Reviewer for pointing out that the substrate's dielectric constant should be considered. According to the results of Fig. 3 in Rosenstein and Ya Shapiro, PRB 103, 224517 (2021), with dielectric constant in the range of 30~10000, the superconductivity enhancement effect is weakly dependent on the dielectric constant. According to the literature [S. M. Khetre, et al., Adv. Appl. Sci. Res. 2(4), 503-511 (2011); M. Idrees, et al., Acta Materialia 59, 1338-1345 (2011)], the dielectric constant of LFO is in the range of 1500 to 3500, within the insensitive range. We appreciate the reviewer for pointing it out and making our discussion more rigid.

Following the Reviewer's suggestions, we analyzed the spectral weight of replica band. In Fig. R6(a), we show our analysis on the intensity ratio, which is 0.33 for γ band, and 0.52 for β band. The intensity ratio should be the same for different bands, while here the difference between γ band and β band is probably because the choice of background is not ideal. In FeSe/LFO, the replica band separation is smaller than that in FeSe/STO, and thus the replica band γ' and spectral weight from band β partially overlap (black arrow), making the determination of background more difficult than that in FeSe/STO. In Fig. R6, by choosing three different backgrounds, the determined intensity ratio of γ band is different. On the other hand, the intensity ratio of the β band shows a minor influence from the background, this is because that the replica band β' is well separated from other bands, and the intensity ratio of β'/β should represent the EPC strength more reliably. In FeSe/STO, the replica band β' is hardly observed, while the clear observation of β' in the raw data indicate a larger EPC in FeSe/LFO. The determined intensity ratio between 0.40-0.52, is clearly larger than that in FeSe/STO (~0.2). This is qualitatively consistent with the larger EPC strength determined according to the blue shift of the energy separation relative to the phonon energy.

Fig. R6 Analysis of the spectral weight ratio of the replica bands using three different backgrounds. The spectral weight ratios of the replica band are illustrated below each analysis.

Following the referee's suggestions, in the revised manuscript, we have cited the above references, and added discussions on the dielectric constant of the substrates. Besides, we have also mentioned the intensity ratio analysis in the main text, and added Fig. R6 and corresponding discussions into the revised Supplementary Information.

4) Related to the above point, the authors write about the cross-interfacial e-ph coupling scenario as if it's a given. However, it remains quite controversial. For example, work from George Sawatzky's group suggests that the replica bands may have nothing to do with e-ph coupling [PRL 120, 237001], while recent work from Kyle Shen's group [arXiv:2102.03397] seems to speak against this proposal. Similarly, the authors should mention the recent FLEX works [PRB 102, 180501(R); PRB 103, 144504], which arrive at quite different conclusions on whether forward scattering can be combined with spin fluctuations in this system to enhance T_c . These works show that the cross-interfacial e-p coupling scenario is far from accepted, and the reader should, at the very least, be made aware of these issues. Later in the paper, the authors also elude performing RIXS measurements to study the spin excitations; these measurements have in fact, been done, see Pellicciari et al., arXiv:2008.09618.

Reply: We thank the reviewer for pointing out these 5 papers, which are closely relevant to our work.

1. Work from George Sawatzky's group mentioned the possibility that replica bands originated from the coupling between photoelectrons and interfacial phonons [F. Li and G. A. Sawatzky, PRL 120, 237001 (2018)], which does not have any blueshift. However, our experiments showed that the energy separation between original and replica bands is higher than the surface phonon energy and is independent on the incident photon energy, which

supports the intrinsic interfacial EPC mechanism [Q. Song, T. L. Yu, et al., Nat. Commun. 10, 758 (2019) and this work]. Moreover, in the recent work from Kyle Shen's group [B. D. Faeth, et al., Phys. Rev. Lett. 127, 016803 (2021)], experiments with changing photon energy in a larger range also nicely demonstrated a discrepancy with the extrinsic EPC.

2. In the two FLEX calculation papers, PRB 102, 180501(R) theoretically calculated the effects on the FeSe superconducting symmetry from spin fluctuation and interfacial EPC, while PRB 103, 144504 pointed out that with proper relative strength, the cooperation between EPI and spin fluctuation results in the superconductivity enhancement from 46K to 65K. These results are highly related to the mechanism of superconductivity enhancement and should be cited.

3. Pellicciari et al. [Nat. Commun. 12, 3122 (2021)] measured the different spin excitations in both single layer and bulk FeSe materials, indicating that single layer FeSe/STO with electron doping has a magnetic excitation different from bulk FeSe. This is an important and related work. On the other hand, we would like to clarify that what we intend to study is the spin excitation of LFO in FeSe/LFO, which have not been studied. Whether the spin excitation of LFO takes extra effects on superconductivity of the FeSe film has not yet been studied, for which RIXS could be an effective mean to study FeSe/LFO interface in the future.

In the revised manuscript, we have cited the above literatures, and rewritten the introduction part to include the current controversies and experimental progress. We have also checked our manuscript thoroughly, and avoiding writing about the cross-interfacial e-ph coupling scenario as if it's a given, and making the discussion more rigorous.

5) It would be valuable to the community to compare the current results against the reference FeSe/STO interface (which superconducts) and FeS/STO interface, which does not [see K. Shigekawa et al, PNAS 116, 2470 (2019)]. These two systems also have different distances between the FeSe layer and the STO substrate. How do these compare?

Reply: We thank the Reviewer for the helpful comments. Although FeS/STO and FeSe/STO also show different interlayer distance, the major difference between them is the different correlation effect in FeSe and FeS, which results in the different superconducting properties. Compared with FeSe, FeS has a larger band width and correspondingly a weaker correlation, and interfacial EPC cannot effectively take effects for enhancing the superconductivity in the weakly correlated system according to the recent work by Rosenstein and Ya Shapiro [PRB 103, 224517 (2021)]. Their calculation suggests the interfacial distance and correlation strength are two parameters that both influence the superconductivity [see Fig. 2 in Rosenstein and Ya Shapiro, PRB 103, 224517 (2021)]. Here in our FeSe/LFO system, the electron correlation is similar to FeSe/STO from the observation of its electronic structure. This provide a clean example to study the effect on superconductivity from interlayer distance. Moreover, the FeS-STO bond length is also larger than that of optimized FeSe-STO, rendering a weaker interfacial EPC.

Following the Reviewer's suggestions, we have added the above discussion and the comparison between single layer FeS/STO and FeSe/STO in our revised manuscript.

6) Throughout the paper, the authors refer to Ref. 26 as having performed

"approximation free quantum Monte Carlo simulations." I strongly object to this description. When calculating the T_c enhancements, the authors of that paper introduced an electronic pairing interaction by hand. Moreover, they considered finite-size lattices, which approximate the thermodynamic limit. Both of these are approximations. I suggest that the authors remove instances of "approximation free" throughout the paper and instead refer to them as QMC calculations.

Reply: We thank the reviewer for pointing this out. We have removed the expression of "approximation free" in our revised manuscript.

Reviewer #3 (Remarks to the Author):

In this manuscript, the authors engineered a new FeSe/LaFeO₃ interface to study the role of interfacial electron-phonon coupling (EPC) in superconductivity enhancement. Angle-resolved photoemission spectroscopy and mutual inductance measurements reveal superconducting pairing at a higher temperature (~80 K) than that which was previously observed in FeSe/SrTiO₃ (~65 K). The authors suggests that the shortened interfacial bond length leads to stronger EPC, and this result may be used in future studies to engineer new interfaces that enhance superconductivity.

I find that the paper is well-motivated and the researchers have completed a thorough set of measurements, which leads to a reasonably complete and coherent story about superconductivity at the FeSe/LaFeO₃ interface.

However, I believe the results could be explained more clearly. Specifically, in the part about the band splitting in the ARPES data shown in Fig. 2. It would also be important to expand on it and clarify the conclusion regarding superconductivity enhancement through the heterostructure design.

Reply: We thank the reviewer for summarizing our result and the positive evaluation of our work. We also thank the constructive comments that make our discussion more rigorous.

We agree that the band splitting should be explained more clearly in Fig. 2. In our last-submitted manuscript, we called it a "splitting", which is indeed confusing; in fact, there are multiple nearly-degenerate bands near EF around Gamma and M due to multi-orbital nature, and the clear identification of the nearly degenerate bands indicate nice data quality. In heavily electron-doped FeSe, the multiband nature around Gamma and M has been reported in M. Yi, et al. Nat. Commun. 6, 7777 (2015), C.H.P Wen, et al., Nat. Commun. 7, 10840 (2016), etc. The degeneracy of the two bands around M can be influenced by strain. For example, in the single layer FeSe film with enhanced tensile strain and larger in-plane lattice of 3.99Å, the two electron pockets in the M point will become more elliptical and the non-degeneracy will be more obvious [R. Peng, et al., PRL 112, 107001 (2014)]. FeSe/LFO/STO has an in-plane lattice same to STO, so it has the same degree of non-degeneracy as that in FeSe/STO. The clear identification of the nearly degenerate bands is brought by small energy and momentum broadening of the spectra, which reflects the high quality of our sample.

In the revised manuscript, we have avoided using the word "splitting" to avoid confusion, and expand our discussion on the clear identification of nearly degenerate bands,

following the referee's suggestions.

Some other comments and questions:

1. In Figure 1, what are some of the theoretical explanations for the origin of the band splitting at the Γ and M points? Presumably it is related to the electron-phonon coupling, which would be an interesting finding - therefore, it is worthwhile to explain this feature more thoroughly in the text.

Reply: We thank the Reviewer for pointing this out. Figure 1 shows the STEM results about the sample structure and not related with the band splitting, and the reviewer should be talking about the band splitting in Fig. 2. The splitting of α and β bands, and the splitting of γ_1 and γ_2 bands are the multi-orbital features of heavily electron-doped FeSe without interfacial EPC [M. Yi, et al., Nat. Commun. 6, 7777 (2015), C. H. P. Wen, et al., Nat. Commun. 7, 10840 (2016)], while the replica bands located at about 88meV below these main bands are originated from the interfacial electron phonon coupling (EPC). More specifically, the EPC corresponds to the interaction between the 80 meV phonon in LFO and the electrons in FeSe [Ref. 39].

In the revised manuscript, we have explained this feature more thoroughly in the description text about Fig. 2.

2. Furthermore, it is unclear why this splitting is an indication of a high-quality sample. E.g. could such splitting be an indicator that the analyzer has sufficient energy and momentum resolution.

Reply: We agree that the width of peaks in MDC and EDC can be influenced by the energy and momentum resolution of the analyzer used in experiments; however, in our setup, the resolution of the analyzer is high enough but is not the most critical limitation for resolving the nearly degenerate bands. As shown in Fig. R7, two nearly degenerate bands can be observed in #1 and #2, but is blurred in #3, while these data were measured with the same analyzer setup, and the width reflects the impurity scattering in these three different samples (see Fig. R7), which will broaden the spectral function in the energy and momentum.

In the revised manuscript, we have added Fig. R7 into the revised manuscript, and added "Due to the small energy/momentum separation between them, the clear identification of each band under the same experimental setup reflect high quality of the samples with minimal impurity scattering" in the main text.

Fig. R7 The band structure in Gamma point of different single layer FeSe samples measured

by the same ARPES with same setup. Sample #1, #2 and #3 correspond to the 1uc-FeSe/LFO (T=6K) presented in our manuscript, a high-quality 1uc-FeSe/STO (T=7K), and a moderate-quality 1uc-FeSe/STO (T=8K), respectively. As the red arrows indicated, in both sample #1 and #2, the two nearly-degenerate bands can be clearly observed, while in the sample #3, the two nearly-degenerate bands cannot be distinguished.

3. Have there been any DFT calculations done on this structure? If so, do the calculations predict the observed band splitting?

Reply: If the reviewer means the splitting of α/β bands and γ_1/γ_2 bands, former polarization dependent ARPES experiments suggested that it is caused by the multi-orbital features of FeSe [M. Yi, et al., Nat. Commun. 6, 7777 (2015) , Y. Zhang, et al., PRL 117, 117001 (2016)]. LDA calculated band structure showing multiple bands were shown in Nekrasov, I. A. & Sadovskii, M. V., JETP Lett. 93, 166–169 (2011) and the calculation was compared with experiments in M. Yi, et al., Nat. Commun. 6, 7777 (2015).

If the reviewer means the splitting between the replica bands and main bands, there are several theoretical calculations, such as Z.-X. Li, et al., PRB 100, 241101(R) (2019) and L. Rademaker, et al., PRB 103, 144504 (2021).

We have cited the corresponding references in the revised manuscript to make the discussion clearer following the referee's suggestions.

4. It would be good to see the RHEED pattern for the 1.5uc sample as a point of comparison to the 1uc sample.

Reply: We thank the reviewer for the suggestion. However, our FeSe chamber is not equipped with RHEED, and we have not measured RHEED for the 1uc and 1.5uc samples. However, according to the in-situ STM data which show atomically flat terraces, and the cross-sectional STEM data which show coherent strain of both 1uc and the 2nd uc FeSe, we expect that the RHEED of 1uc and 1.5uc samples should show two dimensional streaks with the same in-plane lattice parameter.

5. Is it possible to improve the interconnection of the FeSe films by changing the growth condition e.g. growing at a higher temperature or laser power? Some discussion would be good to see.

Reply: We thank the reviewer for the helpful suggestion. As FeSe-FeOx interface is sharp from the STEM studies, obtaining high quality LFO with pure FeOx termination could be important for increasing the interconnection of FeSe. Changing the temperature for LFO growth and tuning the laser power could tune the termination, and might be beneficial to increasing the coverage ratio of single layer FeSe film. Besides, using oxide-MBE technique to grow LFO film in a layer by layer mode could achieve a better control of the terminated plane of LFO, which could be helpful for obtaining FeSe/LFO interface with much higher quality. Further studies are certainly needed to be conducted in the future.

Following the referee's suggestions, we have added the above discussion to the revised manuscript.

Once these issues listed above are resolved I can recommend this manuscript for publication.

Reply: We appreciate the reviewer's recommendation and helpful suggestions.

REVIEWERS' COMMENTS

Reviewer #1 (Remarks to the Author):

The authors provided answers to my questions. Let me go over them one by one.

1. The explanation is clear. It is unfortunate that 1 uc FeSe cannot be measured but that is the limit of the ex situ measurements.
2. I miss the fact that with only 6 uc of LFO, the LFO film is strained to STO. Just mentioning this fact along with a statement that strain is not a factor should be good enough.
3. Well explained.
4. Just stating the fact that Fe and Ti have very similar ionic radii should be enough. As they are similar, shorter bond means stronger bond.
5. STS would be nice but it is not essential.
6. qualitatively accounted by... => qualitatively accounted for by...
7. We learn a lot from failed attempts. Therefore, even providing the available work function of MgO should be helpful. It is still within the work function mismatch scenario.

I believe the authors answered my questions/comments to a satisfactory level. My view is that the importance of this work lies in the fact that they were able to grow 1 uc superconducting FeSe on non-TiO_x surface for the first time. This should be considered a significant advance in the field.

My comments (as well as other reviewers') are mostly on interpretations, which in my view are secondary.

Reviewer #2 (Remarks to the Author):

I have examined the revised manuscript by Song et al., together with the author's responses to the comments raised by myself and the other reviewers. My observations and criticisms have been addressed in the revised manuscript or the author's reply. I also want to thank the authors for engaging with the comments and for the analysis of the replica bands in their sample. This addition will be helpful for further research. Based on the revised manuscript and correspondence with the Reviewers, I recommend publication of the manuscript. As outlined in my initial report, I believe that observing superconductivity in these FeSe/LaFeO₃/SrTiO₃ structures provides an essential contrasting example to FeSe on TiO₂ terminated substrates. In my mind, the discovery of this case suggests several promising and exciting research directions for testing the proposed mechanisms for enhanced pairing in FeSe monolayers. In particular, the coupling to the 80 meV LFO mode suggested by the author's work will provide important clues in our effort to understand the potential role of interfacial e-ph coupling.

Reviewer #3 (Remarks to the Author):

I went over the revised manuscript and the ref. response and find the answers satisfactory. With these changes, I can recommend the manuscript for publication.

The point-by-point response to all the reviewers' comments are listed as follows.

=====

REVIEWERS' COMMENTS

Reviewer #1 (Remarks to the Author):

The authors provided answers to my questions. Let me go over them one by one.

1. The explanation is clear. It is unfortunate that 1 uc FeSe cannot be measured but that is the limit of the ex situ measurements.

Reply: We thank the reviewer for the positive comment and the understanding.

2. I miss the fact that with only 6 uc of LFO, the LFO film is strained to STO. Just mentioning this fact along with a statement that strain is not a factor should be good enough.

Reply: We thank the reviewer's helpful suggestion. In the revised manuscript, along with the fact, we have added "the strain effect on FeSe in FeSe/LFO/STO is identical to the well-studied FeSe/STO".

3. Well explained.

Reply: We thank the reviewer for the positive comment.

4. Just stating the fact that Fe and Ti have very similar ionic radii should be enough. As they are similar, shorter bond means stronger bond.

Reply: We thank the reviewer's helpful suggestion, and we have added the statement on the similar ionic radii of Fe and Ti.

5. STS would be nice but it is not essential.

Reply: We thank the reviewer for the understanding.

6. qualitatively accounted by... => qualitatively accounted for by...

Reply: We thank the reviewer for pointing out this grammatical mistake and we have corrected it in the revised manuscript.

7. We learn a lot from failed attempts. Therefore, even providing the available work function of MgO should be helpful. It is still within the work function mismatch scenario.

Reply: We thank the reviewer for the positive comment. We have also added the available work function of MgO in the revised manuscript.

I believe the authors answered my questions/comments to a satisfactory level. My view is that the importance of this work lies in the fact that they were able to grow 1 uc superconducting FeSe on non-TiOx surface for the first time. This should be considered a significant advance in the field.

My comments (as well as other reviewers') are mostly on interpretations, which in my view are secondary.

Reply: We thank the reviewer for pointing out our work as “a significant advance in the field”. We also thank the reviewer for all the helpful suggestions, which helped us to improve our manuscript a lot.

Reviewer #2 (Remarks to the Author):

I have examined the revised manuscript by Song et al., together with the author's responses to the comments raised by myself and the other reviewers. My observations and criticisms have been addressed in the revised manuscript or the author's reply. I also want to thank the authors for engaging with the comments and for the analysis of the replica bands in their sample. This addition will be helpful for further research. Based on the revised manuscript and correspondence with the Reviewers, I recommend publication of the manuscript. As outlined in my initial report, I believe that observing superconductivity in these FeSe/LaFeO₃/SrTiO₃ structures provides an essential contrasting example to FeSe on TiO₂ terminated substrates. In my mind, the discovery of this case suggests several promising and exciting research directions for testing the proposed mechanisms for enhanced pairing in FeSe monolayers. In particular, the coupling to the 80 meV LFO mode suggested by the author's work will provide important clues in our effort to understand the potential role of interfacial e-ph coupling.

Reply: We thank the reviewer for his/her positive evaluation and recommendation of this work.

Reviewer #3 (Remarks to the Author):

I went over the revised manuscript and the ref. response and find the answers satisfactory. With these changes, I can recommend the manuscript for publication.

Reply: We appreciate the reviewer's recommendation.